# Analysis of Surface Properties of Nickel Alloy Elements Exposed to Impulse Shot Peening with the Use of Positron Annihilation

**DOI:** 10.3390/ma14237328

**Published:** 2021-11-30

**Authors:** Agnieszka Skoczylas, Kazimierz Zaleski, Radosław Zaleski, Marek Gorgol

**Affiliations:** 1Department of Production Engineering, Faculty of Mechanical Engineering, Lublin University of Technology, 36 Nadbystrzycka, 20-618 Lublin, Poland; k.zaleski@pollub.pl; 2Physics and Computer Science, Institute of Physics, Department of Materials Physics, Faculty of Mathematics, Maria Curie-Sklodowska University, Maria Curie-Sklodowskiej Sq. 1, 20-031 Lublin, Poland; radek@zaleski.umcs.pl (R.Z.); marek.gorgol@mail.umcs.pl (M.G.)

**Keywords:** Inconel 718, impulse shot peening, surface roughness, microhardness surface layer, positron annihilation lifetime, ANOVA analysis of variance

## Abstract

The paper presents the results of experimental studies on the impact of impulse shot peening parameters on surface roughness (Sa, Sz, Sp, Sv), surface layer microhardness, and the mean positron lifetime (τ_mean_). In the study, samples made of the Inconel 718 nickel alloy were subjected to impulse shot peening on an originally designed stand. The variable factors of the experiment included the impact energy, the diameter of the peening element, and the number of impacts per unit area. The impulse shot peening resulted in changes in the surface structure and an increase in surface layer microhardness. After the application of impulse shot peening, the analyzed roughness parameters increased in relation to post-milling values. An increase in microhardness was obtained, i.e., from 27 HV 0.05 to 108 HV 0.05 at the surface, while the maximum increase the microhardness occur at the depth from 0.04 mm to 0.08 mm. The changes in the physical properties of the surface layer were accompanied by an increase in the mean positron lifetime τ_mean_. This is probably related to the increased positron annihilation in point defects. In the case of small surface deformations, the increase in microhardness was accompanied by a much lower increase in τ_mean_, which may indicate a different course of changes in the defect structure consisting mainly in modification of the dislocation system. The dependent variables were subjected to ANOVA analysis of variance (it was one-factor analysis), and the effect of independent variables was evaluated using post-hoc tests (Tukey test)**.**

## 1. Introduction

Shot peening is a finishing method, which consists in hitting the treated surface with small balls or cut wire shot. As a result of the process, the geometrical surface structure is altered, the surface layer is hardened, and compressive residual stresses are generated [1,2,3]. The surface layer formed during the shot peening process, and in particular compressive residual stresses, increase the fatigue strength and life of elements subjected to this process [4,5,6]. Shot peening may also increase the wear resistance in the elements [7,8]. The beneficial effect of shot peening on the fatigue life has also been observed in elements subjected to the wear process after shot peening [9]. The effect of shot peening on corrosion resistance is unclear. Investigations of the effect of very intensive shot peening on the intergranular corrosion of 304H steel yielded a negative result [10]. In turn, the shot peening process considerably reduced the corrosion of 904L steel welding joints [11]. Shot peening exerts an effect on the adhesive properties of surfaces, which may influence the strength of adhesive joints in elements subjected to shot peening [12].

Shot peening is mainly applied to elements that are exposed to variable loads during use. This process is applied in elements made of a variety of metal alloys, especially steel, titanium alloys, and aluminum alloys. There are relatively few publications on the application of the shot peening process to nickel alloys.

The results of studies on the effect of broaching and shot peening on the microstructure and properties of the surface layer of Inconel 718 gas turbine discs were presented in ref. [13]. Broaching resulted in generation of tensile residual stresses and induced material cracking and plucking. The shot peening process contributed to generation of compressive residual stresses at a depth of approx. 300 µm and an increase in the microhardness of the surface layer. Investigations of shot peening and vibro-peening of a nickel-based superalloy showed higher compressive residual stresses and a greater increase in surface microhardness after the shot peening process than after vibro-peening [14]. As demonstrated by Ortiz, the value and depth of compressive residual stresses in the surface layer of the C-2000 nickel alloy depend on the medium used in the shot peening process and surface mechanical attrition treatment [15]. As a result of shot-peening objects made of the Inconel 718 nickel alloy, compressive stresses arise and microstructure changes [16,17]. Shot peening also improves the fatigue properties of nickel alloys. It has been found that shot peening of the Inconel 718 alloy can result in a 2–20-fold increase in the fatigue life depending on the conditions of the process [18]. The fatigue strength largely depends on the condition of the edges of the tested objects. The research on shot peening of the edges of Inconel 718 samples showed an increase in fatigue strength only in certain conditions [19]. Shot peening increases the fatigue strength of elements made of nickel alloys working at elevated temperatures [20]. Nickel alloys are treated with laser shot peening as well [21].

The process of shot peening of additively manufactured nickel alloys has been investigated as well. The analysis of the nucleation and propagation of fatigue cracks in non-peened and peened Inconel 718 nickel alloy samples produced with the additive manufacturing technique was described in ref. [22]. Shot peening is successfully used to process stents made of the intermetallic Nitinol (NiTi) alloy in order to increase their wear resistance [23].

The results of shot peening are assessed with various methods. The most common approaches include measurements of the shot peening intensity with Almen plates, assessment of the coverage of the shot-peened surface, analysis of the geometric surface structure, assessment of changes in the microstructure and microhardness of the surface layer, and analysis of the residual stress distribution. Analyses of shot-peened surface layers using positron annihilation-based techniques, which are successfully used to detect defects in metals, have also been carried out [24]. Previous studies have shown that annihilation techniques can be successfully used to analyze the surface layer of shot- peened unalloyed steel, titanium, and aluminum alloys [25], and stainless steel [26]. Additionally, nickel superalloys similar to that analyzed in this study have been investigated with these techniques [27,28,29,30,31,32], and interesting results were reported. Contrary to the other methods mentioned above, positron annihilation provides information about the structure of sample defects (both their size and concentration) at the atomic level. It facilitates correlation of the macroscopic properties of tested materials with their microscopic structure and, consequently, elucidates processes taking place during shot peening. This ensures more effective designs of effective finishing processes.

The literature review indicates that there are only few studies on the physical properties of the surface layer and functional properties of shot-peened nickel alloy elements. There are also no studies on the properties of shot-peened nickel alloys carried out with annihilation techniques or on the influence of technological parameters of impulse shot peening on the properties of the surface layer of nickel alloy elements. Therefore, it seems advisable to determine the impact of the technological parameters of impulse shot peening on surface layer properties with the use of positron annihilation. The aim of the study was to identify technological parameters of impulse shot peening that contribute to low surface roughness and a high increase in microhardness. In turn, detection of changes in the sample defect structure accompanying the improvement in sample properties may elucidate processes involved in impulse shot peening, which may contribute to further improvement of the procedure.

## 2. Materials and Methods

The experiment was carried out using samples made of nickel alloy from the HRSA (Heat Resistant Super Alloys) group, i.e., Inconel 718. This material is a precipitation-hardened nickel alloy with excellent corrosion resistance in many environments, creep resistance, susceptibility to forging and casting, as well as good weldability [33,34]. With its favorable properties, the Inconel 718 nickel alloy is used in such machine parts as turbines, discs, shafts, compressor blades, exhaust outlets, and combustion chambers [35]. It constitutes over half the mass of a conventional turbojet engine. The drawback of the nickel alloy is its low resistance to frictional wear, which nevertheless can be eliminated by surface processing [35]. Table 1 presents the chemical composition and properties of the Inconel 718 nickel alloy.

Figure 1 shows the experimental design consisting of a set of variables and output data.

The Inconel 718 nickel alloy samples were milled prior to the impulse shot peening process. The six-blade milling head with an outer diameter of *D_g_* = 40 mm was equipped with round cemented carbide plates covered with TiAl coating. The following technological parameters were employed in the milling process: depth of cut—*a_p_* = 0.5 mm, cutting speed—*v_c_* = 40 m/min, and feed per tooth—*f_z_*= 0.08 mm/blade. Abundant cooling with the Mobilcut cooling-lubricant fluid was applied in the process.

Impulse shot peening was performed on an originally designed shot peening stand (Figure 2a). The indentations were produced in a regular manner (Figure 2c), i.e., one next to another at a step of “*x*”. On the Figure 2b is presented the schema of the device. During shot peening, the work surface of the workpiece (2) was subjected to the hits of the beater (5) of a known mass, raised to the height “h” by the cam (8). The beater with a ball with a diameter *d* (3), which is an interchangeable element, moves while operating in the ball guides (4). The CNC table, on which the workpiece is attached, performs a feed movement, and the speed of this movement may be controlled with a guide screw (9) and the stepper motor (10, 12). The speed of the feed motion affects the value of the shot peening density (number of impacts per unit area). The impact energy *E* depends on the mass of the beater, mass of the weight, and the height of the peening ball drop. 

The impulse shot peening process was applied at variable technological parameters, which were selected on the basis of preliminary research, presented in Table 2. The peening density is shown in Equation (1), where *j* is the number of impacts per unit area:(1)$$j=\frac{1}{x^{2}}\left[{\mathrm{mm}}^{-2}\right]$$

The T800RC 120–140 Hommel-Etamic device (Jenoptik, Villingen-Schwenningen, Germany) was used for the analysis of the 3D surface topography. The area of the scanned surface was 4.0 mm × 4.0 mm. The Hommel Map Basic v. 6.2 software (Jenoptik, Villingen-Schwenningen, Germany) was used for determination of the parameters of 3D surface roughness.

Microhardness was measured with the Vickers method on diagonal sections after standard treatment in accordance with the EN-ISO 6507-1:2018 standard. An LM 700at microhardness tester (Leco, St. Joseph, Michigan, USA) was used with an indenter load of 50 g (HV 0.05).

Positron annihilation lifetime spectroscopy (PALS) measurements were conducted using a digital positron lifetime spectrometer with an Agilent U1065A digitizer (Acqiris, currently supported by Keysight Technologies, Santa Rosa, CA, USA) (sampling rate of 4 GS/s,) and a specialized program [36] for determination of the interval between signals with amplitudes corresponding to energy of radiation emitted during positron formation (1274 keV) and annihilation (511 keV). The pulses were generated by two scintillation detectors equipped with BaF_2_ scintillators, placed in the immediate vicinity of the samples. Each of the detectors was used to detect both radiation quanta with energy of 1274 keV and 511 keV. Positrons were produced by ^22^NaCl with an activity of 0.3 MBq deposited on an 8 μm thick Kapton envelope between two identical samples, which were mounted in a dedicated holder. To avoid the coincidence of the 511–511 keV annihilation quanta, antiparallel and emitted along a single line, the sample with the positron source was placed in such a way that no line passed through the source and both scintillators simultaneously. The positron lifetime spectrum for each sample was collected with the total number of counts of approx. 2.2 × 10^7^. The chosen measurements were repeated, placing the positron source in other areas in the sample to verify whether the result was dependent on the position of the source on a potentially non-homogeneous surface.

The preliminary analysis of the PALS measurement results was carried out with the use of the MELT software (Université de Genève, Genève, Switzerland) [37]. It revealed the presence of two dominant components in the spectra, but this method of analysis yields large statistical dispersions (Figure 3). The main analysis of the data performed using the PALSfit program (Technical University of Denmark, Roskilde, Denmark) [38] showed that, in addition to these components, it is necessary to assume the existence of a long-life component with a lifetime of approx. 1.75 ns and a negligible intensity of <0.2% in order to obtain a good fit. This was necessary due to the very low background level (approx. 35 counts/channel at 8000 channels covering a timebase of 50 ns). During the analysis, a correction nt for annihilation in the source envelope with a lifetime of 382 ps and a contribution of 14.44% determined in the Positron Fraction program [39] was assumed. The resolution curve was approximated by a single Gaussian with FWHM ≈ 206 ps.

The effect of the shot peening conditions on the results of surface roughness parameters Sa and Sz (the most frequently analyzed parameters in engineering practice), an increase in microhardness ΔHV 0.05 (caused by impulse shot peening compared to the post-milling value), and the mean positron lifetime τ_mean_ was determined using the analysis of variance (ANOVA) performed in the Statistica version 13 program. Before the ANOVA analysis, the normality of data distribution was examined with the Shapiro-Wilk test and the homogeneity of variance was assessed using the Levene test. The significance level α = 0.05 was assumed in all the analyses. The results of the statistical analysis of variance F were compared with the critical value F_α_ for the adopted significance level and degrees of freedom. The analysis of the effect of the independent variables was verified by means of post-hoc tests (Tukey test). Detailed results of the analyses are presented in the Appendix A.

## 3. Results and Discussion

### 3.1. Surface Roughness

Figure 4 shows the effect of the impact energy on the analyzed 3D surface roughness parameters. The increase in the impact energy was accompanied by a clear rise Sa (arithmetical mean height of the surface), Sz (maximum height of the surface) (Figure 4a), Sp (maximum peak height of the surface), and Sv (maximum pit height of the surface) (Figure 4b). This is related to the formation of deeper depressions on the shot-peened surface. The application of high impact energies caused substantial deformations of post-milling micro-roughness (increased value of the Sp) and formation of numerous depressions (increased Sv). In terms of the impact energy used in the experiment, the analyzed parameters of the 3D surface roughness increased in comparison with these values obtained after the milling process.

At the constant impact energy and shot peening ball diameter, the growth the distance between the indentations reduced the degree of coverage. This contributed to uneven deformation of the shot-peened surface and, consequently, increased the roughness parameters (Figure 5). The effect of the impact density *j* on surface roughness parameters was more noticeable at the value of *j* = 6 ÷ 25 mm^−2^. 

In the process of impulse shot peening with the use of a ball with a small diameter, the contact surface of the ball with the workpiece is relatively small, hence the more intense plastic-elastic deformations result in an increase in surface roughness. In turn, the impact of a ball with a larger diameter produced shallower indentations on the surface, which caused more intensive micro-roughness smoothing after milling and a decrease in surface roughness parameters Sa, Sz, Sp, and Sv (Figure 6). At a ball diameter of *d* = 12.45 mm, the analyzed parameters of 3D surface roughness reached the values similar or lower than those obtained for the milled surface.

The statistical analysis confirmed the significant effect of the technological parameters on the values of the 3D roughness parameters.

### 3.2. Surface Topography

Figure 7 shows the topography and 3D surface roughness parameters prior to impulse shot peening. The surface topography after milling was characterized by an even distribution of micro-roughness with clearly visible elevations and depressions resulting from the geometric-kinematic mapping of the tool in the workpiece. The elevations and depressions represent similar proportions in the total surface profile. The surface topography should be classified as a directed structure.

The surface topography was altered after the impulse shot peening (Figure 8). Numerous indentations induced by the impact of the ball were visible on the shot-peened surface. The depth of the indentations depended on the technological parameters of the process (the maximum Sv was obtained for *d* = 6.00 mm, *j* = 11 mm^−2^, *E* = 240 mJ). They were arranged in successive rows corresponding to the course of the impulse shot peening process. The analysis of the topography (Figure 8e,f) revealed that the deformation of the sample surface was more complete at a higher value of the shot peening density of *j* = 44 mm^−2^ than at *j* = 6 mm^−2^, which was reflected in the over two-fold reduction of the surface roughness. The impulse-shot peening process resulted in changes in the skewness parameter Ssk, which suggests that the material was concentrated around the peaks of the profile; thus, it can be regarded as a good bearing surface [40]. The lower values of the skewness parameter allow an assumption of a greater ability to transfer contact loads and lower tribological wear of the surface in the presence of a lubricant [41].

### 3.3. Microhardness

The impulse shot peening process resulted in changes in the surface microhardness (Figure 9). Impulse shot peening contributed to growth in the density of dislocations, which propagated and were halted when they encountered other dislocations, grain boundaries, or precipitates. The “blockage” of the arising dislocations contributed to an increase in the microhardness parameters. The increase in microhardness close to the surface ranged from 27 HV 0.05 to 108 HV 0.05, and the maximum changes in microhardness reached the depth from 0.04 mm to 0.08 mm.

The effect of the impulse shot peening parameters on the microhardness of the Inconel 718 nickel alloy surface is shown in Figure 10, Figure 11 and Figure 12.

The rising impact energy resulted in a greater increase in the microhardness of the surface layers and in the depth of the hardened layer (Figure 10). The growth the impact energy over *E* = 120 mJ did not produce a significant increase in the surface layer microhardness, but there were evident differences in the depth of the changes.

The decrease in the shot peening density *j*, corresponding to the increase distance between traces *x*, contributed to reduction of the microhardness of the surface layer (Figure 11). 

An increase in the diameter of the shot peening ball *d* contributed to enlargement of the impact-induced indentation surface area. This led to reduction of the concentration of energy transferred to the workpiece, thereby lowering the microhardness of the surface layer of the Inconel 718 samples (Figure 12). Noteworthy, while approximately parallel shifts of microhardness distributions with a similar slope were observed in the case of changes in *E* and *j*, the distributions became flatter with the increase in *d*. This indicates that the increase in the ball diameter is accompanied by a decrease in the microhardness at the surface and the rise in the depth of the surface layer hardening.

The statistical analysis confirmed the significant effect of the technological parameters on the values of the ΔHV 0.05 microhardness increase (Appendix A).

### 3.4. Positron Lifetime Spectroscopy

The procedure of averaging the results of all measurements shows the mean values of the lifetime of the two dominant components in the positron lifetime spectra of <τ_1_> = 137 ps and <τ_2_> = 197 ps. These values are too close to allow determination of the parameters (lifetime and intensity) of each component with a acceptable accuracy. Consequently, the lifetimes in the ranges: τ_1_ = 103 ÷ 160 ps and τ_2_ = 165 ÷ 246 ps are observed. This impedes unequivocal interpretation of the origin of the components. Moreover, it should be expected that they may have several sources of origin with similar lifetimes. The lifetime of the first component for most of the samples is too long to originate from annihilation in undefected material, where lifetimes in the range of 110 ÷ 120 ps [42] are expected, which should additionally be reduced due to positron trapping [43]. Bulk lifetimes in the range of 146 ÷ 166 ps, which are clearly too large, would rather result from the lifetimes of the first component and intensity of the second component in the range of 9 ÷ 79%. Therefore, it is most probable that the first component originates from the annihilation of positrons trapped in the dislocations. These are shallow traps with positron binding energy below 0.1 eV. Positrons can relatively easily leave the traps and then annihilate in adjacent point defects thus contributing to the second component [44]. This, in turn, may modify the lifetime of the first component. The lifetime of the second component indicates that it may be a result of positron localization in Ni, Fe, or Cr monovacancies. Incoherent δ phase precipitates (Ni_3_Nb) or niobium carbide at the grain boundaries may also contribute to this interphase annihilation component. Due to the afore mentioned low accuracy of determination of the lifetime values and intensities of each of the components, the mean positron lifetime (τ_mean_) is the most reliable parameter, which can be determined using the formula, Equation (2):(2)$$\tau_{\mathrm{mean}}=\frac{\tau_{1}I_{1}+\tau_{2}I_{2}}{I_{1}+I_{2}}$$
where: τ_i_ and I_i_ are the lifetime and intensity of the ith component of the positron lifetime spectrum. A low value of τ_mean_ indicates a dominant role of dislocations in positron trapping, and its increase is associated with a greater relative contribution of point defects and, probably, grain boundaries.

To compare τ_mean_ with microhardness measurements, the weighted mean of microhardness HV_w_ was determined for each sample, Equation (3).
(3)$$HV_{w}=\frac{\sum_{i}\Delta N_{i}HV0.05_{i}}{\sum_{i}\Delta N_{i}}$$
where: ΔN_i_ is the number of positrons, determined based on the positron implantation profile [45], annihilating in the ith layer, for which microhardness is HV0.05_i_ (Figure 10, Figure 11 and Figure 12). This allows taking into account the uneven contribution of the individual material layers for which microhardness was measured to τ_mean_ (analogous to the residual stresses in ref. [46]).

The comparison of τ_mean_ and HV_w_ for different impact energy values is shown in Figure 13. In the untreated sample, τ_mean_ ≈ 157 ps corresponded to HV_w_ ≈ 470. The growth in the impact energy to 60 mJ was accompanied by an approximately linear increase to τ_mean_ ≈ 168 ps and HV_w_ ≈ 540. This evidences an increase in the concentration of point defects. Importantly, τ_mean_ indicates the ratio of defects to dislocations; nevertheless, a decrease in the concentration of dislocations caused by the shot peening seems unlikely. In turn, changes in their structure cannot be excluded, i.e., the dislocations became shorter, which resulted in faster de-trapping of positrons propagating along the dislocation line and trapping in point defects often accompanying dislocations [47]. Above energy of 60 mJ, τ_mean_ had a constant value within measurement errors, and the increase in the HV_w_ value was inconsiderable. This indicates saturation of the alterations in the samples. The upper sensitivity limit for PALS is known, i.e., the absence of an increase in the mean lifetime despite an increase in the defect concentration when all positrons are trapped in a given type of defect. However, the absence of an increase in HV_w_ suggests that this was not the case here.

The increase in the shot peening density resulted in a rapid rise in HV_w_ to 510 at *j* = 6.25 mm^−2^ followed by a much milder HV_w_(j) relationship reaching 540 at the maximum peening density used in the study (Figure 14). 

In this case, the changes in the τ_mean_ value did not follow the same trend as HV_w_, but exhibited a nearly linear increase in the entire range with no initial sharp rise. Noteworthy, in the case of the same scale of the mean τ_mean_ and HV_w_ axes (0.12 ps/HV_w_), the rate of changes in both values in the range over *j* = 6 mm^−2^ was similar. During the investigation of the effect of shot peening density with PALS, the selected shot peening parameters (*E* = 40 mJ, *d* = 10.00 mm) were chosen. This was made to avoid the influence of surface non-homogeneity which caused the dependence of the result on the location of the positron source, and was the case especially when smaller balls and higher energy values were used (Appendix A). Choosing these particular parameters resulted in a smaller surface deformation at a single impact (Table 3).

Most probably, little point defects (observed at larger deformations) were created in the sample in this case, but the dislocations were modified (e.g., shortened), which had a positive effect on microhardness but was not detected by PALS. Similar differences between τ_mean_ and HV_w_ are shown in Figure 13 and Figure 15 for the smallest surface deformations. 

Interestingly, there is no analogical dependence on the energy density (quotient of impact energy and the indentation area), which indicates that that not all energy is transferred to non-elastic deformations when different shot peening parameters are used.

The change in the diameter of the peening ball had a weak effect on HV_w_ (Figure 15), which at larger ball sizes (6–13 mm) rised slightly in the range of 540–560 along the decrease in the ball diameter (i.e., an increase in transferred energy per unit area). This was related to the increase in the slope of the HV 0.05 distributions accompanying the decrease in the size of the ball (Figure 12), which compensated for their increasingly shallower penetration into the sample. This trend seemed to be halted in the case of a ball with a diameter of 3.95 mm, for which the HV_w_ value declined slightly to 530. The τ_mean_ dependence agreed well with the dependence of HV_w_(d) at the same scale for both quantities as that for the dependence on impact energy. An exception was the ball with the diameter of 12.45 mm, for which τ_mean_ was clearly lower than the expected value resulting from HV_w_. This discrepancy may have had the same background as the dependence on the shot peening density, i.e., the formation of defects that were not detected by PALS at small surface deformations induced by a single impact. In this case, the strong effect may have resulted from the low curvature of the indentation, which resulted in smaller deformations than in the case of high curvature of the overlapping indentations.

The statistical analysis confirmed the significant effect of the technological parameters on the values of the mean positron lifetimes.

## 4. Conclusions

The impulse shot peening process resulted in changes in the geometrical structure of the surface. In comparison with the pre-treatment, shot peening contributed to a decrease in the value of the skewness parameter Ssk. This allows an assumption of lower tribological wear in the presence of the lubricant on the surface of the Inconel 718 subjected to the impulse shot peening procedure. This is confirmed by the presence of numerous depressions on the surface, which may serve as lubrication pockets. The occurrence on the surface of the lubrication pockets allows to reduce the coefficient of friction during the cooperation of two elements with each other. Unfortunately, the other surface roughness parameters (Sa, Sz, Sv, and Sp) increased in relation to the post-milling values.

Impulse shot peening carried out at *E* = 180 mJ, *j* = 11 mm^−2^, and *d* = 12.45 mm yields lower values of surface roughness parameters Sa, Sz, Sp, and Sv than the values of post-milling parameters and a large depth of strengthening (z = 80 μm). This suggests an increase in the wear resistance of the surface of Inconel 718 workpiece subjected to shot peening at these parameters, which should therefore be considered the best among those used in the research for the process.

The positron lifetimes indicate annihilation of two groups of positrons: trapped in dislocations and in various types of point defects (vacancies, grain boundaries). The changes in the ratio of the concentration of defects of both types can be estimated basing on the changes in the mean lifetime. There is a good correlation between the mean lifetime and the weighted mean microhardness HV_w_ (with the positron implantation profile as the weight) for HV_w_ > 500. This implies that an increase in microhardness is associated with the growth in the probability of positron trapping in point defects. This may result from the increase in their concentration and shortening of the dislocation line, which increases the probability of de-trapping from dislocations and trapping in point defects. In the case of the HV_w_ value in the range of 470–500, the increase in the mean lifetime is lower than 0.12 ps/HV_w_ observed for higher microhardness values. This is probably due to the different modification of the defect structure responsible for the increase in microhardness with smaller deformations per impact, which results in less intense formation of point defects. It mainly consists in modification of dislocations, i.e., shortening there of or formation of a larger number of mutually blocking dislocations.

## Figures and Tables

**Figure 1 materials-14-07328-f001:**
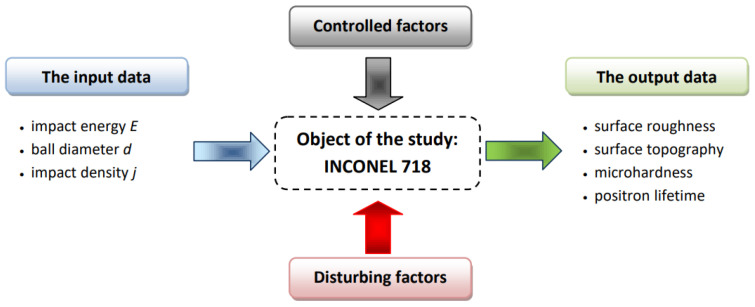
Scheme of the experiment design with lists of input parameters and output data. The scheme shows the influence of controlled and disturbing factors.

**Figure 2 materials-14-07328-f002:**
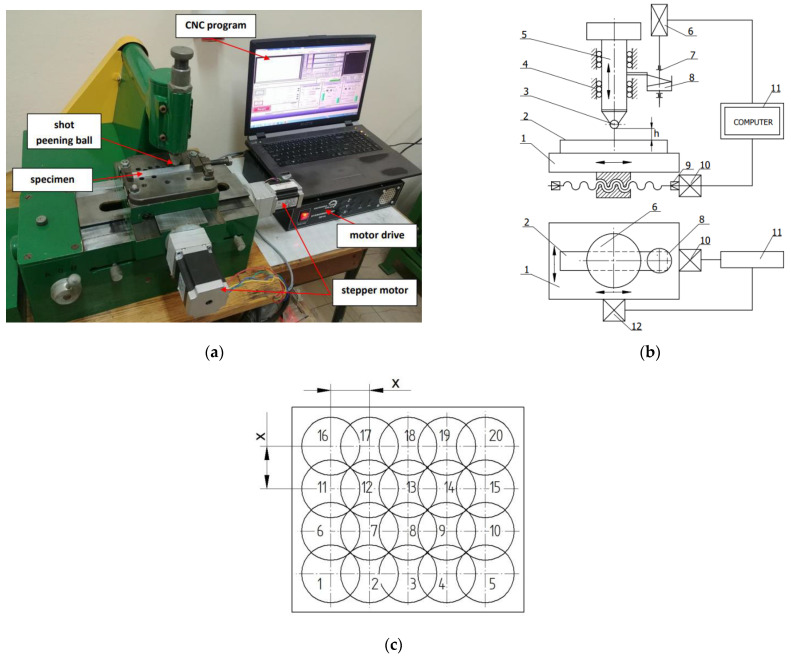
Originally designed impulse shot peening stand (**a**), schema of a stand for impulse shot peening (**b**) 1—CNC table, 2—workpiece, 3—ball, 4—ball guides, 5—beater, 6—engine, 7—shaft, 8—cam, 9—guide screw, 10—stepper motor, 11—computer, 12—stepper motor, and scheme of indentations on the peened surface (**c**).

**Figure 3 materials-14-07328-f003:**
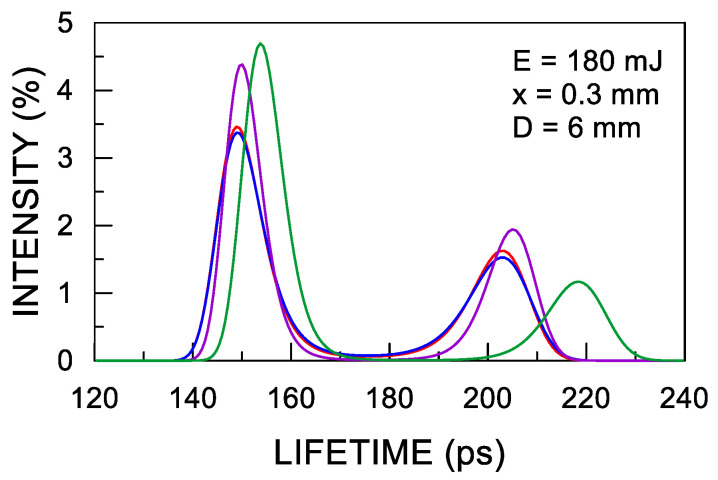
Typical positron lifetime distributions determined from four spectra measured in the same conditions for a selected sample with shot peening parameters shown in the figure.

**Figure 4 materials-14-07328-f004:**
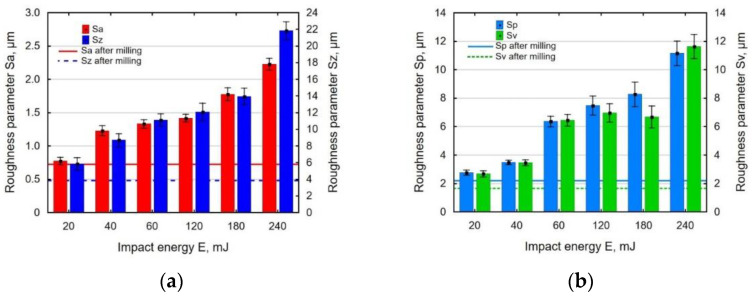
Effect of impact energy *E* on surface roughness parameters: Sa and Sz (**a**); Sp and Sv (**b**) (*j* = 11 mm^−2^, *d* = 6.00 mm).

**Figure 5 materials-14-07328-f005:**
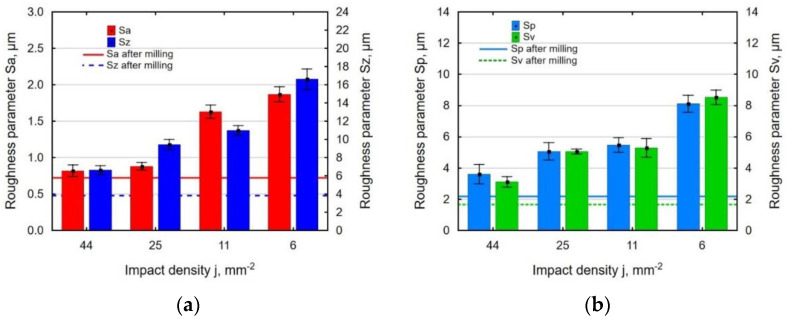
Effect of impact density *j* on surface roughness parameters: Sa and Sz (**a**); Sp and Sv (**b**) (*E* = 40 mJ, *d* = 10.00 mm).

**Figure 6 materials-14-07328-f006:**
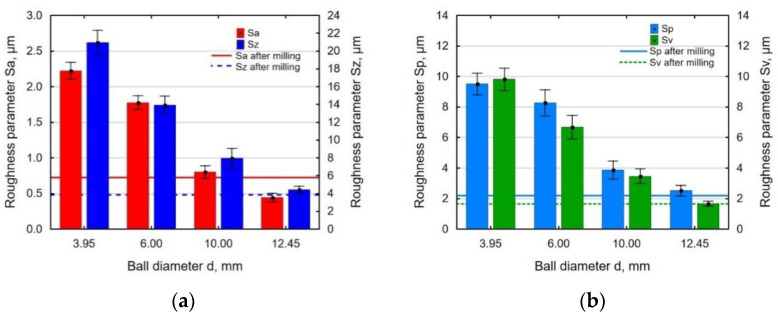
Effect of peening ball diameter *d* on surface roughness parameters: Sa and Sz (**a**); Sp and Sv (**b**) (*E* = 180 mJ, *j* = 11 mm^−2^).

**Figure 7 materials-14-07328-f007:**
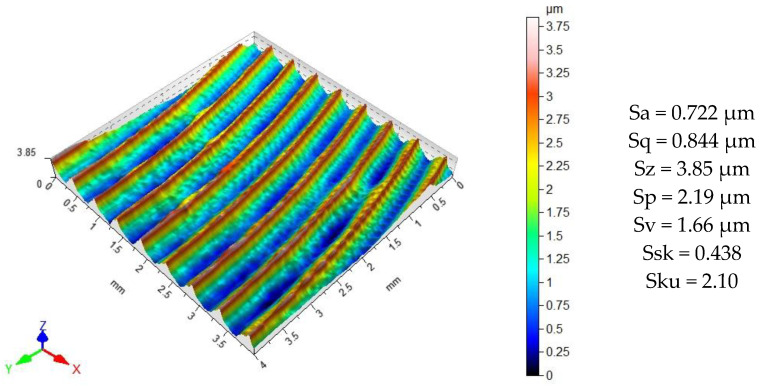
Topography and 3D surface parameters after milling.

**Figure 8 materials-14-07328-f008:**
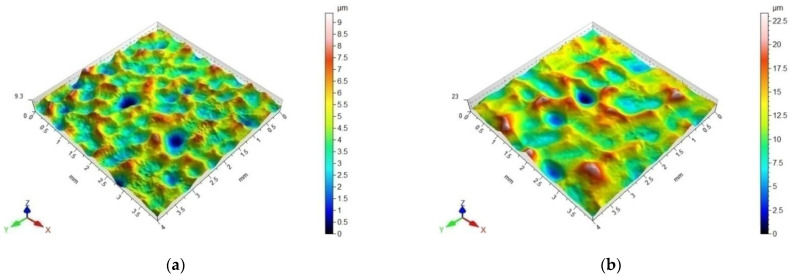
Topography of the surface of Inconel 718 samples after impulse shot peening: (**a**) *E* = 20 mJ, *d* = 6.00 mm, *j* = 11 mm^−2^; (Sa = 1.05 µm Sz = 9.39 µm Sp = 4.89 µm, Sv = 4.50 µm, Ssk = 0.059), (**b**) *E* = 240 mJ, *d*= 6.00 mm, *j* = 11 mm^−2^; (Sa = 2.44 µm Sz = 23.30 µm Sp = 11.40 µm, Sv = 11.90 µm, Ssk = 0.202), (**c**) *E* = 180 mJ, *d*= 3.95 mm, *j* = 11 mm^−2^; (Sa = 1.80 µm Sz = 15.90 µm Sp = 8.47 µm, Sv = 7.45 µm, Ssk = 0.258), (**d**) *E* = 180 mJ, *d* = 12.45 mm, *j* = 11 mm^−2^; (Sa = 0.41 µm Sz = 3.68 µm Sp = 1.96 µm, Sv = 1.73 µm, Ssk = 0.166), (**e**) *E* =40 mJ *d* = 10 mm, *j* = 44 mm^−2^; (Sa = 0.76 µm Sz = 8.38 µm Sp = 4.82 µm, Sv = 3.58 µm, Ssk = 0.135), (**f**) *E* = 40 mJ, *d* = 10 mm, *j* = 6 mm^−2^; (Sa = 1.90 µm Sz = 16.30 µm Sp = 7.69 µm, Sv = 8.62 µm, Ssk = 0,092).

**Figure 9 materials-14-07328-f009:**
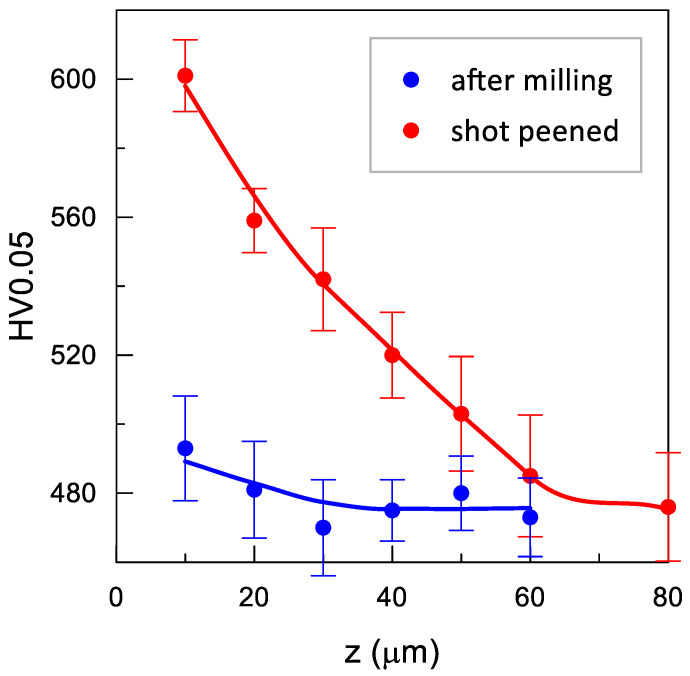
Distribution of the microhardness of the Inconel 718 nickel alloy surface before and after the impulse shot peening process with parameters *E* = 180 mJ, *d* = 3.95 mm, *j* = 11 mm^−2^.

**Figure 10 materials-14-07328-f010:**
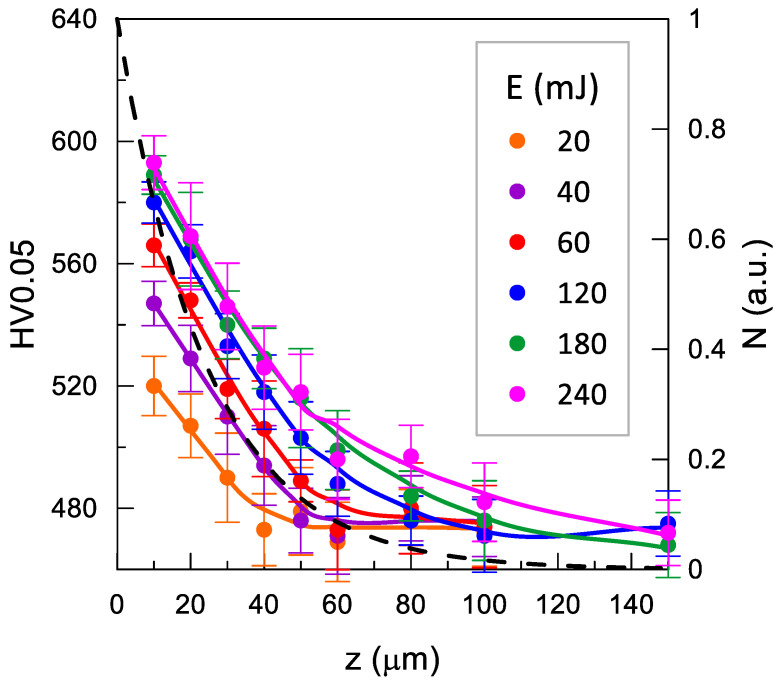
Effect of impact energy *E* on microhardness distribution (*j* = 11 mm^−2^, *d* = 6.00 mm). The dashed line shows the positron implantation profile (N); the solid lines are only an eye-guide.

**Figure 11 materials-14-07328-f011:**
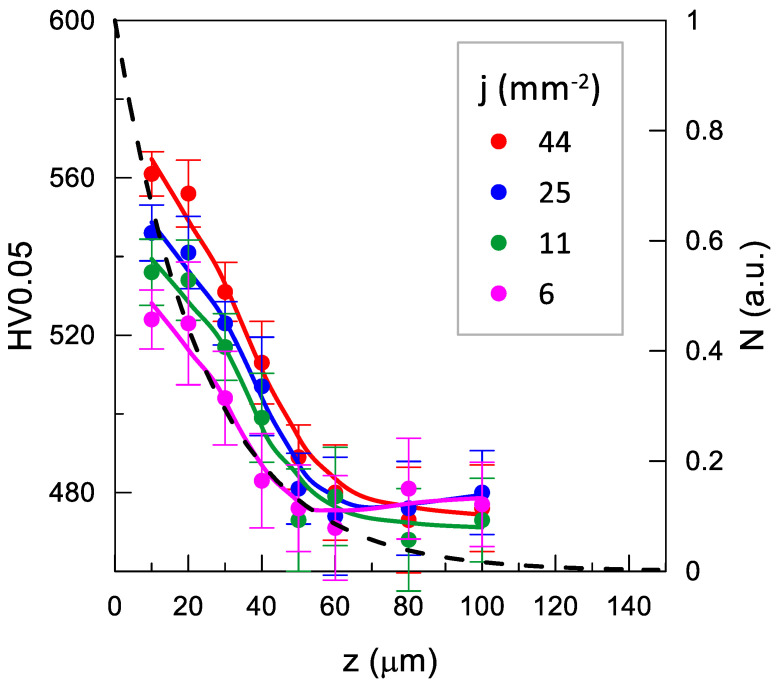
Effect of shot peening density *j* on microhardness distribution (*E* = 40 mJ, *d* = 10 mm). The dashed line shows the positron implantation profile (N); the solid lines are only an eye-guide.

**Figure 12 materials-14-07328-f012:**
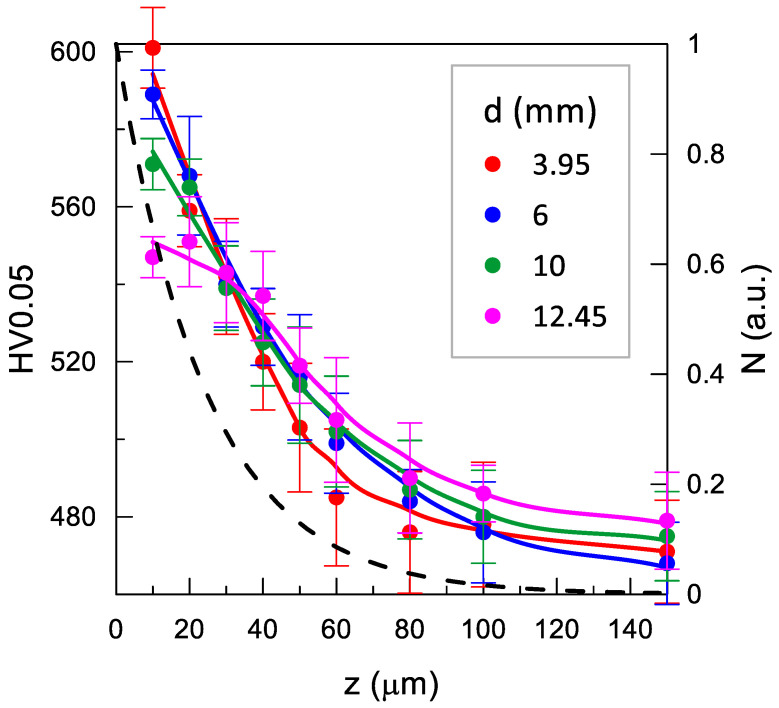
Effect of ball diameter *d* on microhardness distribution (*E* = 180 mJ, *j* = 11 mm^−2^). The dashed line shows the positron implantation profile (N); the solid lines are only an eye-guide.

**Figure 13 materials-14-07328-f013:**
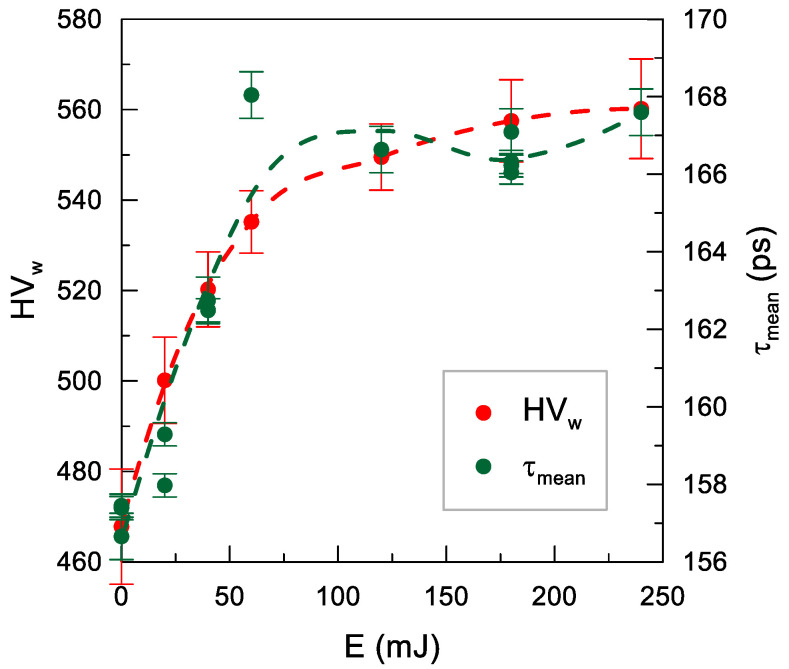
Dependence of the weighted mean of microhardness HV_w_ and the mean positron lifetime τ_mean_ on the impact energy *E* (*j* = 11 mm^−2^, *d* = 6.00 mm). The dashed lines are only an eye-guide.

**Figure 14 materials-14-07328-f014:**
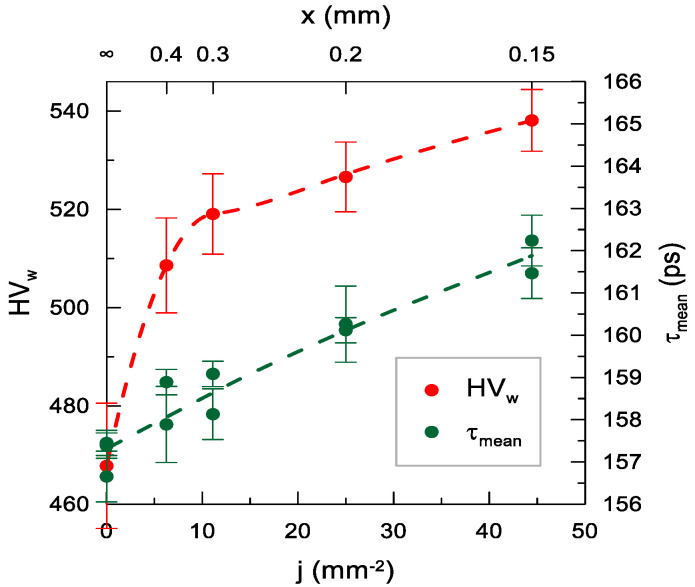
Dependence of the weighted mean of microhardness HV_w_ and the mean positron lifetime τ_mean_ on the peening density *j* (*E* = 40 mJ, *d* = 10.00 mm). The dashed lines are only an eye-guide.

**Figure 15 materials-14-07328-f015:**
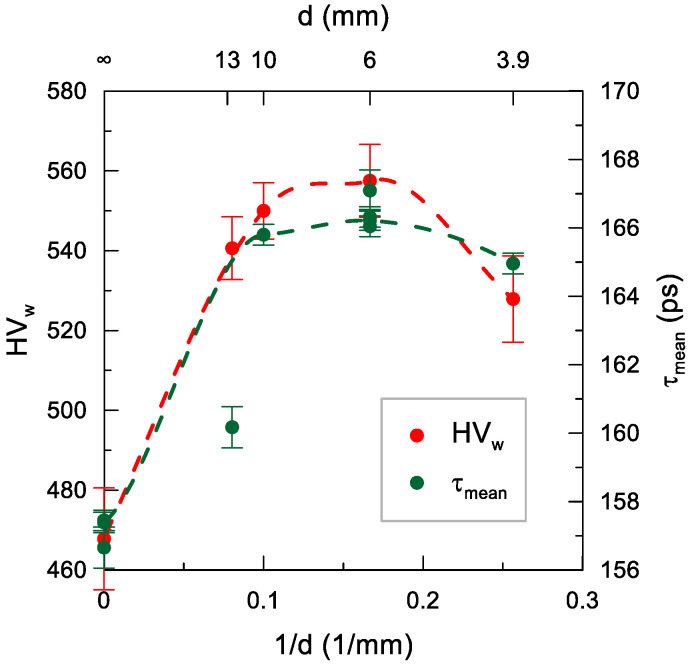
Dependence of the weighted mean of microhardness HV_w_ and the mean positron lifetime τ_mean_ on the ball diameter d (*E* = 180 mJ, *j* = 11 mm^−2^). The dashed lines are only an eye-guide.

**Table 1 materials-14-07328-t001:** Chemical composition and properties of the Inconel 718 nickel alloy [34].

Chemical Composition (%)						
Cr	Fe	Nb (+Ta)	Mo	Ti	Co	
17–21	11.16–22.50	4.75–5.50	2.80–3.30	0.65–1.15	1	
Al	Si	Mn	Cu	C	B	Ni
0.2–0.8	0.35	0.35	0.3	0.08	0.006	balance
Tensile strength (MPa)				1400		
Yield point (MPa)				864		
Young’s modulus (GPa)				205		

**Table 2 materials-14-07328-t002:** Technological parameters of Inconel 718 impulse shot peening.

No.	Impact Energy, *E*, mJ	Ball Diameter *d*, mm	Distance between Traces *x*, mm	Peening Density *j*, mm^−2^
1	20	6.00	0.30	11
2	40			
3	60			
4	120			
5	180			
6	240			
7	40	10.00	0.15	44
8			0.20	25
9			0.30	11
10			0.40	6
11	180	3.95	0.30	11
12		10.00		
13		12.45		

**Table 3 materials-14-07328-t003:** Effect of impulse shot peening conditions on the deformation depth and energy density (quotient of the impact energy and the indentation area).

Impact Energy *E*, mJ	Ball Diameter *d*, mm	Peening Density *j*, mm^−2^	Indentation Diameter *d_o_*, mm	Maximum Depth of a Single Deformation *h*, mm	Energy Density g_E_, mJ/mm^2^
20	6.00	11	0.259	0.006	378.6
40			0.562	0.026	160.8
60			0.664	0.031	172.8
120			0.906	0.069	185.7
180			0.960	0.077	248.0
240			0.981	0.081	316.7
180	3.95		0.670	0.057	509.2
	10.00		0.971	0.047	242.5
	12.45		0.975	0.038	240.4
40	10.00	44	0.702	0.025	103.1
		25			
		11			
		6			

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
