# Peer review of "Analysis of Surface Properties of Nickel Alloy Elements Exposed to Impulse Shot Peening with the Use of Positron Annihilation"

_materials, 2021, doi:10.3390/ma14237328_

Round 1

Reviewer 1 Report

  1. English should be improved. In particular, there are too many of the same words side by side in many places. More synonyms should be used, for example for the word ‘increased’. It is not clear in what context the word ‘element’ is used in many places next to ‘nickel’. The first thing that comes to mind is a chemical element, but something else is mentioned, apparently. The phase ‘the Inconel 718 nickel alloy’ is like ‘steel consisting iron’. The phrase ‘the microhardness changes 19 reached a range from 0.04 mm to 0.08 mm’ is not correct. There are many other troubles with the text.
  2. Table 1 needs to be made more readable. Why Hardness [HB] of the initial material is presented if microhardness was measured after surface processing?
  3. Caption for Figure 1 should be improved.
  4. Column headings should be fully indicated in Tables 2 and 4.
  5. ‘Technological parameters of Inconel 718 impulse shot peening’ should be justified.
  6. May be ‘the most frequently analyzed parameters in engineering practice’ are Ra and Rz , but not Sa and Sz?
  7. Table 3 looks more like a picture than a table.
  8. The ‘Discussion’ section is missing, which is inappropriate. The authors have reported individual results, but did not analyze them and compare them with those already published by other researchers. Accordingly, it is impossible to judge the correctness of the research findings.
  9. The conclusions are too long and are presented in the format of a laboratory study, but not a scientific article. Only the obtained data are reported, but there is no justification why it happened that way and what follows from this.

Reviewer 2 Report

Improving surface properties through surface treatment processes such as shot peening is the current research focus of high-performance surface processing. The article explores the effects of pulse peening process parameters such as impact energy, shot peening element diameter and number of impacts per unit area on surface roughness, surface micro-hardness and average positron lifetime. It has important guiding significance for the application of pulse peening process in surface strengthening. The following questions need the author's confirmation:

  1. Figure 2 shows the device diagram. However, it is difficult to see the specific structure, shot peening control part and processing principle of the device in Figure 2. It is recommended to add necessary explanations and a schematic diagram of the device structure to make it easier to understand.
  2. The purpose of subtitle “4 positron lifetime spectroscopy” in this paper is not clearly described in detail. Is it used to characterize micro-structure? If yes, it is recommended to add relevant technical background and application in related fields.
  3. The roughness in Figure. 4 and Figure. 5 shows a different increasing trend with the increase of peening energy, so it is suggested to add relevant introduction. In particular, the roughness parameter Sv in Figure. 4(b) shows an inconsistent variation trend when the impact energy is 180, correlation analysis is suggested.
  4. Reference [14]does not appear in the citation of the article, and the author needs to check it carefully.
  5. The three-line table in Table 1 is incorrect. The energy parameter Eof sample groups 11-13 in Table 2 was not given.
  6. Many details need to be further sorted out, for example “In the process of impulse shot peening with the use of a ball with a small diameter (d= 3.95÷00 mm)” Why is the diameter of ball “d” expressed as a fraction?
  7. References have inconsistent paragraph formats, such as reference[3] andreference [33]. Reference [35] and references [40] are published in 2009 and 2003 respectively, but they are still shown as “in Polish”. Besides some references are too old.
  8. Some of the content of the introduction is not closely related to the text, for example “Eddy currents can also be used for measurements of residual stresses and evaluation of the micro-structure of shot-peened Inconel 718 samples .”

Based on above analysis, it is considered that the research content of this paper is innovative and has certain reference value for engineering application, so it is recommended to be published in this journal after minor repairs.

Round 2

Reviewer 1 Report

The manuscript could be published if editors deem it appropriate.